# The Presence of Human Herpesvirus 6 in the Brain in Health and Disease

**DOI:** 10.3390/biom10111520

**Published:** 2020-11-06

**Authors:** Gabriel Santpere, Marco Telford, Pol Andrés-Benito, Arcadi Navarro, Isidre Ferrer

**Affiliations:** 1Neurogenomics Group, Research Programme on Biomedical Informatics (GRIB), Hospital del Mar Medical Research Institute (IMIM), DCEXS, Universitat Pompeu Fabra, 08003 Barcelona, Catalonia, Spain; 2Institute of Evolutionary Biology (UPF-CSIC), Departament de Ciències Experimentals i la Salut, Universitat Pompeu Fabra, PRBB, 08003 Barcelona, Catalonia, Spain; marco.telford@upf.edu (M.T.); arcadi.navarro@upf.edu (A.N.); 3Centro de Investigación Biomédica en Red de Enfermedades Neurodegenerativas (CIBERNED), 28031 Madrid, Spain; pol.andres.benito@gmail.com; 4Catalan Institution of Research and Advanced Studies (ICREA), Passeig de Lluís Companys 23, 08010 Barcelona, Spain; 5Centre for Genomic Regulation (CRG), Barcelona Institute of Science and Technology (BIST), Carrer del Dr. Aiguader 88, 08003 Barcelona, Spain; 6Barcelonaβeta Brain Research Center (BBRC), Pasqual Maragall Foundation, Wellington 30, 08005 Barcelona, Spain; 7Department of Pathology and Experimental Therapeutics, University of Barcelona, Hospitalet de Llobregat, 08907 Barcelona, Spain; 8Bellvitge University Hospital, IDIBELL (Bellvitge Biomedical Research Centre), Hospitalet de Llobregat, 08908 Barcelona, Spain

**Keywords:** human herpesvirus 6, brain, encephalitis, multiple sclerosis, Alzheimer’s disease, immunohistochemistry, epilepsy

## Abstract

The human herpesvirus 6 (HHV-6) -A and -B are two dsDNA beta-herpesviruses infecting almost the entire worldwide population. These viruses have been implicated in multiple neurological conditions in individuals of various ages and immunological status, including encephalitis, epilepsy, and febrile seizures. HHV-6s have also been suggested as playing a role in the etiology of neurodegenerative diseases such as multiple sclerosis and Alzheimer’s disease. The apparent robustness of these suggested associations is contingent on the accuracy of HHV-6 detection in the nervous system. The effort of more than three decades of researching HHV-6 in the brain has yielded numerous observations, albeit using variable technical approaches in terms of tissue preservation, detection techniques, sample sizes, brain regions, and comorbidities. In this review, we aimed to summarize current knowledge about the entry routes and direct presence of HHV-6 in the brain parenchyma at the level of DNA, RNA, proteins, and specific cell types, in healthy subjects and in those with neurological conditions. We also discuss recent findings related to the presence of HHV-6 in the brains of patients with Alzheimer’s disease in light of available evidence.

## 1. Introduction

The human herpesvirus 6 (HHV-6) -A and -B are two globally dispersed dsDNA beta-herpesvirus infecting almost the whole worldwide population [1,2,3]. These two viruses were until recently classified as a single entity due to their high nucleotide identity, whereas they actually present multiple differences at the level of epidemiology, biology, including their primary receptor, and immunological profiles [4,5,6]. After its initial isolation in 1986 from AIDS patients in USA [7], reference sequences for HHV-6A and HHV-6B were established: U1102 from Uganda [8] and Z29 from Zaire [9], respectively. New technological advances in targeted genomic capture coupled to next-generation sequencing have greatly expanded the catalogue of HHV-6 complete genome sequences to few hundreds [10] and have revealed major geographic patterns among isolates, large differences between HHV-6A and HHV-6B levels of genetic diversity, and the need of using a more representative reference strain for HHV-6B [11,12].

Primary infection with HHV-6 mainly occurs through saliva via nasopharynx and olfactory entry routes early in life [13]. Upon infection, telomere-like repeats encoded in the HHV-6 genome allows the virus to integrate in the sub-telomeric region of human chromosomes and establish latency [14,15]. HHV-6 was initially described as a lymphotropic virus [7], but although it replicates most efficiency in CD4+ cells in vitro and in vivo [16,17], multiple in vivo and in vitro observations have shown HHV-6 in a plethora of cell types, including epithelial cells, fibroblast, astrocytes, oligodendrocytes, and neurons [18,19,20,21,22].

In around 1% of the population, the HHV-6 genome is present in all nucleated cells of the body as the result of a previous infection and integration in progenitors’ germinal cells. This form of HHV-6 viruses was first reported in the 1990s [23] and is called inherited chromosomally integrated HHV-6 (iciHHV-6). Integrated viruses can be reactivated with certain drugs or in immunocompromised individuals [24], and this event has been associated with angina pectoris and diverse inflammatory diseases [12,25]. Moreover, individuals with iciHHV-6 can incidentally render high viral load in the nervous system, which is relevant for the diagnose of diseases caused by active primary HHV-6 infections that have been horizontally transmitted [26]. Nevertheless, this review will focus mostly on the presence of HHV-6 in the brain originated from an exogenous, horizontal, infection and the subsequent invasion of the nervous system.

While infection by HHV-6 is generally asymptomatic, HHV-6B is the etiological factor for *exanthema subitum* (roseola infantum) [27] which most often produces a non-life-threatening fever and skin rash. Other conditions have been associated to the acute infection or reactivation in immunocompromised individuals, including AIDS encephalopathy, several types of cancer, demyelinating diseases such as multiple sclerosis (MS) and progressive multifocal leukoencephalopathy (PML), chronic fatigue syndrome, febrile seizures, multiple transplant complications, heart, lung and liver diseases, cognitive disfunction, epilepsy, and encephalitis [22,25,28,29,30,31]. Recently, some studies have implicated the infection by HHV-6 in the pathogenesis of Alzheimer’s disease (AD) [32,33]. This has raised a heated debate as well as a lot of effort from multiple research groups to replicate, confirm or challenge this putatively fundamental observation on a disease with such an elevated prevalence world-wide and social burden. The debate around the suggested contribution of HHV-6 in AD requires to step back to summarize what we known so far about the presence of HHV-6 in the brain under multiple neurological conditions and detection methods, its cell tropism in vivo and in vitro and suggested effects.

## 2. HHV-6 Can Infect Multiple Neural Cell Types 

The capacity of HHV-6 to infect neural cells in vitro, albeit with less efficiency than lymphocytes, was established in the late 1980s using neuroblastoma, glioblastoma, and embryonic glia cell lines [34,35,36]. The presence of HHV-6 in the brain has been repeatedly reported since the 1990s [37,38] and it is widely admitted in both immunocompromised and immunocompetent individuals. Detection methods used have included PCR, real-time PCR, in situ hybridization, immunohistochemistry, electron microscopy, and more recently, droplet digital PCR (ddPCR) and high-throughput sequencing techniques, such as RNA-seq and DNA-seq. Considering together observations from healthy individuals and from patients of multiple conditions, HHV-6 has been reported in vivo in oligodendrocytes [39], astrocytes [40], glial precursor cells [41], and also in different types of neurons, including cortical and hippocampal neurons [42,43] and Purkinje cells of the cerebellum [22]. 

In this review we have summarized experimental designs, methodologies, and observations from 48 studies analyzing a total of 1393 samples (excluding around 1000 samples from two large-scale studies with undefined sample overlap; [32,44] from multiple areas of the human brain under multiple neurological conditions. For a given condition, the variety of methods used to detect HHV-6, HHV-6 typing, brain regions assessed, ages, clinical heterogeneity and sample sizes, among other factors, makes it challenging to derive strong quantitative estimates, but allow to derive qualitative observations regarding the range of frequency in which a particular neural cell type is found infected by HHV-6. For some conditions, statistically adequate and updated meta-analysis across studies have been recently made available [45,46] and are discussed in the corresponding sections. We devoted special attention to studies reporting the presence of HHV-6 at the protein level, indicating HHV-6 activity in the brain. In Figure 1, we represented the proportion of studies focusing on a given neurological condition that reported immune-reactivity for HHV-6 in each neural or infiltrating cell type, from the data collected and organized in Appendix A.

Of the 48 studies considered in this review, 40 investigated the presence of HHV-6 DNA by means of either PCR, nested PCR, in situ PCR, qPCR, or ddPCR. Only four studies analyzed HHV-6 gene expression using qPCR or RNA-seq, however those studies based on RNA-seq represent the best-powered to date in terms of sample size. A total of 30 out of the 48 studies used histo/cytochemical antigen detection in tissues and cells. The majority of these studies (26/30) used formalin-fixed and paraffin-embedded tissue. We have summarized all reported antibodies used to detect HHV-6 in the nervous system (Table 1). When the information was available in the corresponding publications, or kindly provided by authors or companies upon our queries, the clone of origin is indicated. We also indicate when possible which kind of protein is targeted (early or late in the lytic phase) and which type of HHV-6 is recognized, i.e., type A, type B or both. Studies using immunohistochemistry to detect HHV-6 in the nervous system globally report the results of around 15 different antibodies, some of them obtained from multiple sources. The antibody C3108-103, which recognize the late antigen p101, a tegument protein, specific of HHV-6B and published by Dr. Phillip Pellet et al. in 1993 [47] is the most widely used; in a total of 13 studies. This is followed by the antibody C5 raised against p41, an early nuclear antigen [48] targeted in 10 studies. This antibody is used to recognize both HHV-6 A and B, and in various studies authors used both p101 and p41 antibodies to cover the detection of both types. However, according to the catalogue of Advance Biotechnologies, their version of this p41 antibody derived from the C5 clone apparently lose the capacity to detect both types and that it begun to detect only type A. The same conclusion might apply to the original C5 clone. The third most used antibody recognizes gp116/gp64/gp54 and was produced by Balachandran et al. in the late 1980s [49], which recognizes both A and B variants, and is used in at least 8 studies. The rest of antibodies have been raised against additional antigens (i.e., gp110/60, gH or gp82/gp105), sometime undisclosed, including some cases of non-commercially available antibodies used in studies from more than two decades ago, for which we were not able to retrieve a complete description. These additional antibodies have been used less frequently in a range from 1 to 4 studies. Only one study in this collection analyzed the expression of HHV-6’s U94 latency gene in the brain [50]. In Table 1 reports the cell type recognized by each antibody when the information provided in the paper is univocal. Studies using multiple antibodies but not specifying the results for each independent staining are not considered in this column, but their findings can still be found in Appendix A.

## 3. Detection of HHV-6 in Non-Pathological Samples

The presence of HHV-6 DNA in the brain among individuals without neurological conditions has been repeatedly reported using a variety of probes and primers and PCR techniques. Out of 28 studies scanning for DNA that include non-pathological control samples, 21 show evidence of the presence of HHV-6, with reported prevalence varying substantially (range: 0–100%; median: 26% and range: 4–100%; median: 35%, for all 28 studies and the 21 HHV-6 positive studies, respectively; Figure 2).

Even if the presence of the HHV-6 genome has been repeatedly described in normal brains, it is clear that commonly interrogated HHV-6 antigens, usually derived from early and late lytic genes are only very rarely, or not at all, present in the nervous system of healthy individuals (Appendix A). Occasional findings of HHV-6 positive cells using immunohistochemistry in control samples might also indicate artefacts of the technique, involving unspecific detection of for example amylaceous bodies (Figure 4) or lipofuscin granules. The comparison with sections stained for lipofuscin granules, using periodic acid Schiff, helped discarding a particular neuronal HHV-6 staining in one MS patient [51]. Independent assessment of HHV-6 by PCR can also help to disambiguate certain cases. For instance, in one study the authors found immunopositivity for HHV-6 in cells from healthy subjects, including young individuals [52]. When these samples were tested for the presence of HHV-6, immune-positive cells were found in some PCR-negative samples, casting doubts on the specificity of the staining.

In conclusion common detection of HHV-6 DNA in the brain in immunocompetent individuals is seldom accompanied by the detection of viral proteins, indicating the capacity of HHV-6 to invade the brain and establish a latency program. This latency and ubiquity suggest that HHV-6 is most often a commensal virus in the CNS, where it can have lifelong persistence without causing damage. Reactivation from latency seems to require stimulus from co-infection with certain other viruses such as HHV-7 or activated T-cells, at least in mononuclear cells. Reactivation in the brain, for example in oligodendrocytes, might also require co-infection with other viruses [53], as has been suggested in PML with JCV [54]. 

## 4. Mechanism of Entry of HHV-6 into the Brain Parenchyma

The route used by HHV-6 to infiltrate into the central nervous system is not fully understood, but some evidence has been gathered so far suggests various possible trajectories (Figure 3). It is possible that there exists one or multiple preferred routes of invasion, while alternative routes might be relevant in certain pathological conditions. Comparison of intravenous or intranasal inoculation of HHV-6 in Marmoset established differences in the outcome of the infection, in terms of antibody production and neurological symptoms [6], which suggest the hypothesis that the pathological outcome of HHV-6 infection might be affected by its entry route also in humans.

It has been suggested that HHV-6 might enters into the brain through a brain–blood barrier debilitated by inflammatory pathologies. Observations in favor of this hypothesis include the infection of HHV-6 in vascular endothelial cells and infiltrates of lymphocytes and macrophages adjacent to blood vessels [43,55,56]. HHV-6 infected lymphocytes and adventitial fibroblasts have been observed also in the pia mater [57], which suggest that HHV-6 could also penetrate the meningeal barrier. The blood–CSF barrier established by the choroid plexus is another potential route of entry, which is supported by the presence of HHV-6 antigens in the choroid plexus of certain individuals, for example in one case report of a patient with tuberous sclerosis [58]. Interestingly, expression of HHV-6 in that individual was restricted to infiltrating macrophages and lymphoid cells and absent in neural cells, suggesting that HHV-6 might colonize the brain in the passenger seat of peripheric immune cells travelling from the choroid plexus to the ventricles and from the ventricles to the brain parenchyma. 

The high prevalence of HHV-6 DNA in the brain of healthy subjects, repeatedly reported across decades of studies, implies that other routes must be at play which are non-dependent on the abnormal permeability of brain barriers. High prevalence of HHV-6 DNA has been detected in nasal mucus, olfactory bulb, and tract [13,52] and recent immunohistochemical analysis revealed the expression HHV-6 antigens in multiple cell types in the olfactory bulb and tract, which suggests that HHV-6 could enter the brain through the olfactory pathway. Moreover, it was shown in vitro that HHV-6 can infect a population of glial cells, the olfactory ensheathing cells, which have a role in guiding the axons of olfactory receptor neurons into the CNS [13]. Even if this could represent a common route of entry of HHV-6 among the healthy population, since HHV-6 genome and even protein antigens have been detected among control samples in two studies [52,57], this trajectory could also have direct implications for certain neurological conditions with preferential manifestation in limbic areas, closely connected to the olfactory pathway such as HHV-6 related encephalitis and TLE. Finally, the possible entry of HHV-6 through the eyes has also to be considered either through the optic tract or facial nerves connected to the lacrimal gland. The detection of HHV-6 DNA in the optic tract of ~60% of individuals with no neurological conditions [57] suggests the optic tract as an additional possible entry route of HHV-6. As well as the presence of HHV-6 DNA in tear fluid also open the possibility of a CNS entry through a facial nerve infection [59].

## 5. HHV-6 and Encephalitis

The most acknowledged relationship between HHV6 and neurological damage relates to its role in encephalitis. Encephalitis produced by HHV-6 is rare but it often has devastating sequelae when occurring [60,61]. HHV-6 encephalitis can lead to neurological damage often in the form of ataxia and epilepsy [62,63] and in certain cases it has caused death, both in infants [64,65] and adults [38,66]. It can be the result of a primary infection or viral reactivation, as is more commonly observed in organ transplant recipients, AIDS patients or individuals affected by other immunocompromising pathologies. However, cases of HHV-6 encephalitis have been also reported in immunocompetent adults [67,68]. HHV-6 is found active in around 50% of patients receiving stem-cell or bone marrow transplants [69], and it represents a major cause of post-transplant acute limbic encephalitis in individuals receiving hematopoietic stem cell transplantation [29,43]. The incidence of HHV-6 reactivation is even higher, more than 80%, in recipients of umbilical cord blood transplants [70]. Encephalitis associated to HHV-6 has been described after transplants of other organs, such as liver [71]. Recently, HHV-6 and also Epstein–Barr virus (EBV, or HHV-4) have been described in the brains of patients with Rasmunssen’s Encephalitis (RE) [42]. This condition of unknown etiology is often focalized in one hemisphere and affects mainly infants who sometimes have to undergo hemispherectomy. A viral etiology for RE was suggested since its initial descriptions by Rasmussen himself [72]. 

In multiple studies of HHV-6 associated encephalitis, and also RE, HHV-6 has been detected using immunohistochemistry on top of CSF DNA identification. Of the 20 studies of this heterogenous group of cases with AIDS-related, transplant-related, or idiopathic encephalitis, 15 included immunohistochemical analysis, using a rich repertoire of antibodies directed against at least four HHV-6 proteins mostly in formalin-fixed paraffin-embedded tissue. Taken together, these immunohistochemical analyses demonstrated the presence of multiple HHV-6 antigens preferentially in astrocytes, followed by neurons and less often in oligodendrocytes (Appendix A; Figure 1). Antigens have also been reported in lymphocytes, microglia and macrophages in a few studies, and in endothelial cells in one study, but it is not clear to which extent non-neural cells are specifically accounted for in many of these studies.

Interestingly, the description of neuropathological damage resulting from HHV-6 infection in multiple cases of encephalitis has repeatedly demonstrated different levels of demyelination, from more spread to more local and intense, sometimes carrying severe axonal damage [38,55,68]. Other demyelinating diseases, such as subacute sclerosing panencephalitis or PML, are associated to viral infection (measles and JCV, respectively). Interestingly, colocalization of HHV-6 antigens in oligodendrocytes inclusions typical of PML was found in a case of a patient presenting PML and HHV-6 meningoencephalitis [73]. But the detection of HHV-6 encephalitis/meningoencephalitis is not essential to observe HHV-6 in PML, and it is not uncommon to find HHV-6 DNA specifically in oligodendrocytes around or within demyelinated lesions in PML (Figure 1, Appendix A), often colocalizing with JCV [54,74]. Finally, HHV-6 was recently observed in oligodendrocytes and other cell types in the white matter of the olfactory pathway, accompanied with demyelination, in patients of idiopathic encephalopathy [52]. 

## 6. HHV-6 and Epilepsy

HHV-6 has been related to one of the most common forms of epilepsy, called temporal lobe epilepsy (TLE). Clinically, this refractory manifestation of epilepsy can be in some cases associated to a previous neurological injury caused by febrile seizures, infections, head trauma, or tumors. In some cases, hippocampal sclerosis is present, with atrophy resulting from gliosis and neuronal loss. In some cases of TLE, viral encephalitis, including that caused by HHV-6, has been proposed as a contributing factor to epilepsy. Indeed, HHV-6 and HHV-7 seem to cause one third of cases of the febrile status epilepticus, a condition that increases risk of epilepsy [75]. However, HHV-6 has been reported enriched in cases of TLE with or without an history of encephalitis [40,76,77]. The capacity of both species of HHV-6 to infect astrocytes, albeit with different biological characteristics, (i.e., A establishes a productive infection and B is more prone to latency), and the observation that the presence of HHV-6 can alter the astrocytic uptake of glutamate in primary cultures [77], suggest a possible role for HHV-6 in the impairment of the neural excitatory balance observed in epilepsy. Importantly, studies on the association between HHV-6 and TLE do not consistently report an enrichment [78] and the stratification of the patients according to their history of brain inflammatory conditions (febrile seizures or meningitis/encephalitis) reveals a significant higher number of HHV-6 DNA positive samples among patients presenting those conditions than the rest of TLE patients [76,78]. Moreover, direct comparisons of HHV-6 DNA load between patients with or without hippocampal sclerosis suggest that the former is higher [79]. A recent, comprehensive meta-analysis of the relationship between HHV-6 and refractory TLE [45] incorporates 8 out of the 10 studies described in Appendix A. Focusing only on the detection of DNA, and despite acknowledging a significant publication bias, HHV-6 was found more frequently in TLE surgical brain resections of the temporal lobe relative to tissue from control individuals [45]. This meta-analysis also supports an enrichment on HHV-6 particularly in the subgroup of patients presenting hippocampal sclerosis.

The presence of HHV-6 genomes was analyzed in 10 out of 10 studies on TLE using multiple probes and primers and by means of PCR, nested PCR and qPCR targeting multiple coding regions: U22, U31, U38, U67, and U86 (Appendix A; Figure 2). Astrocytes were HHV-6 positive in all studies performing antigen detection, 6 out of 10, including patients of TLE with no history of encephalitis [40,80]; while another study found HHV-6 in astrocytes exclusively in a patient of TLE also presenting subacute HSV encephalitis, but not in the rest of TLE patients [76]. In the 5 out of 6 studies using IHQ, the authors used antibodies raised against the late core antigen gp116/gp64/gp54, but astrocytes were also detected using anti-p101, anti-gH, and the latency protein U94 (Table 1).

## 7. HHV-6 and Multiple Sclerosis

The relationship between the presence of HHV-6 viruses and MS has been studied for years without reaching unanimous conclusions on its role. MS is an idiopathic auto-immune disease in which myelinated axons are primary targets of a misguided and damaging inflammatory response. The contribution of infectious agents, even as a primary cause, has been proposed for more than hundred years, involving viruses and other pathogens [81,82]. HHV-6 reunited qualities for such a job because it is ubiquitous, neurotropic, and can undergo phases of latency and reactivation, compatible with relapsing/remitting symptoms in MS [83]. Indeed, some serologic analyses has suggested a correlate between anti-HHV-6 antibodies and relapsing/remitting MS episodes [84,85], but the direction of causality of this relationship has yet to be established. From early to more recent studies indicated that DNA levels and antibody titers of HHV-6 were increased in CSF or serum samples in MS [86,87], while some authors reported different results [83,88,89]. 

More compelling evidence of a putative role of HHV-6 in MS came from its presence of its DNA, RNA, and antigens in the brain, colocalizing in plaques. Plaques are lesions commonly found in the white matter of the brain and spinal cord which represent a neuropathological hallmark of MS. These lesions seem to result from the glial reaction to the demyelination processes suffered by axons in the course of the disease [90]. Multiple studies have compared the presence of HHV-6 in the brain between MS and controls yielding sometimes controversial results. These discrepancies can sometimes be clarified by considering which tissue is being analyzed, and what information is provided by each detection method [55]. Looking specifically at the presence of HHV-6 DNA in the brain usually no differences were observed between cases and controls, while the labelling of HHV-6 antigens on the same samples by immunohistochemistry, indicative of an active infection, showed the presence of the virus only in MS brains [51,91]. Immunohistochemistry performed on MS brains has repeatedly and robustly reported HHV-6 antigens in oligodendrocytes, and often also in CNS and peripheral immune system cells: Microglia, lymphocytes, and macrophages [39,51,55,56,91,92]. Whether oligodendrocytes were stained by HHV-6 depended strongly on the proximity to MS demyelinating regions, and increased levels of HHV-6 gene expression where found in plaques compared to normal appearing white matter of the same individuals, although normal white matter in MS patients displayed also higher levels of HHV-6 gene expression than white matter in controls [39]. In addition to the staining of oligodendrocytes, HHV-6 positive neurons have also been described using immunohistochemistry, with a frequency and intensity dependent of their proximity to plaques [91]. However, as previously mentioned, apparent anti-HHV-6 occasional staining of neurons was attributed to lipofuscin granules in another study [51].

Many of those studies detecting HHV-6 in oligodendrocytes in MS mentioned in the above paragraph describe HHV-6B, while fewer detected HHV-6A (Appendix A, Table 1). This is consistent with the higher prevalence of HHV-6B in the general population and the known distribution of -6A and -6B in the adult normal brain [93]. However, we must conciliate such prevalence of HHV-6B in damaged oligodendrocytes in MS with several studies reporting a particular association between HHV-6A and MS, when DNA or antibodies are measured in cell free compartments such as serum, CSF or urine [94,95,96]. A recent large-scale study compared IgG reactivity against the highly divergent coding regions U90-U89 and U11, between HHV-6A and HHV-6B, in the serum and plasma of MS patients and controls, testing a total of around 16,000 individuals [87]. The authors found a positive association of HHV-6A, but not -6B, with MS, and also in cases of pre-MS, before the onset of the disease, in an additional case-control cohort of around 1000 individuals. Moreover, in vitro studies using the MO3.13 oligodendrocyte cell line have shown that HHV-6A and -6B infect differently and have different cytopathic effect in culture in this cell type. HHV-6A appeared in that study to be more damaging and able to establish a more persistent infection compared to -6B, which induced and abortive infection, with no antigen expression and with a viral load fainting over time [97].

The proposed role of HHV-6 in MS includes a direct effect in the etiology of the disease, or an indirect aggravating or bystander effect. The fact that HHV-6 is found more active in the brain of MS, in lesions, but also in normal-appearing white matter and sometime active in a proportion of controls, suggests the possibility that HHV-6 could reactivate at multiple loci in the brain under certain circumstances, e.g., a deregulated immune function, as in MS, and exacerbate the pathology [39,95]. In turn, HHV-6 reactivation could lead to an enhanced inflammatory response, involving astrocytes and microglia and promoting lymphocyte infiltrations; astrocytes, microglia, lymphocytes, and macrophages has also been stained by anti-HHV-6 antibodies in MS [56] (Table 1). Another proposed mechanism relating HHV-6 to MS pathology is molecular mimicry between HHV-6 antigens and myelin producing an auto-immune reaction against the later. It was shown that up to 33% of anti HHV-6 T cell lines derived from MS cases and controls cross-reacted with the myelin basic protein (MBP) [98], one of the components of the protein fraction of myelin. A small peptide is shared between MBP and HHV-6 U24 protein, which raises more CD4+ T-cell cross-reactivity in MS than in controls [99]. This cross-reactivity with myelin is not restricted to CD4+ cells but has also been observed in CD8+ cytotoxic T-cells, which could contribute with direct damage on MBP expressing oligodendrocytes in MS [100]. 

These and other mechanisms by which HHV-6 could enhance neuroinflammation are reviewed in [89], but the possibility of more direct cytopathic effect of HHV-6 on oligodendrocytes is plausible considering the mentioned observations in cell cultures and others [41,97,101]. A recent in vitro study focused on the effect of the latency protein U94 of HHV-6A on oligodendrocyte progenitor cells (OPCs) derived from fetal brain and cultured or transplanted in mice brains [101]. The authors found impaired OPC migration, unrelated to survival or proliferation changes, in those cells expressing U94. These findings suggest a putative in vivo role of HHV-6A in MS by disrupting the function of OPC in their transition to demyelinating loci necessary to repair myelin damage in MS. Even if in this case only HHV-6A was analyzed, members of this team of researchers had already provided evidence in 2004 that both HHV-6A and -6B were able to establish an active infection in cultured glial progenitor cells derived from mid-fetal brain. Evidences of the expression of both early and late viral genes were found in these samples, leading to impaired cell proliferation rates [41]. In consequence, a similar role for HHV-6B in OPC migration remains plausible, despite lack of evidence of HHV-6B being able to carry out an active infection in a oligodendrocyte cell line [97].

An important consideration is that oligodendrocytes are not a homogeneous population of cells. New advances in single-cell technologies have made it possible to better characterize heterogeneity among oligodendrocytes and to compare cases of MS and controls or mice models of the disease. Strikingly, these analyses have revealed disease-specific populations of oligodendrocytes and OPCs displaying themselves immune-protection and antigen presentation functions [102,103]. The relationship between HHV-6 activity and these newly described disease-specific populations of oligodendrocytes and OPC has yet to be explored. 

## 8. HHV-6 and Alzheimer’s Disease

Alzheimer’s disease is neuropathologically characterized by the presence of extracellular Aβ forming diffuse and senile plaques and intracellular inclusions of hyperphosphorylated tau named neurofibrillary tangles. Genetic mutations causing the familial form of AD are associated to genes implicated in amyloid formation, and the cascade hypothesis of Aβ has been the leading framework in AD research, as well as the rationale for investing resources for clinical trials involving drugs dealing with Aβ processing, yet failing so far to prove effective. Neuro-inflammation has been consistently observed in the brains of patients of AD and often discussed as a putative primary pathogenic effect, rather than a reactive event of the pathology. In this line, recent meta-analysis of large-scale genome-wide association analysis (GWAS), involving in total around 74,000 individuals, identified risk loci implicating genes related to the immune response and inflammation [104]. Indeed, many genes implicated in AD risk are non-neuronal and actually highly expressed in microglia [105]. One notorious example is a microglial/macrophage cell surface receptor called TREM2, which has an effect size for the risk of late onset AD similar to that of APOE, the first robustly determined genetic risk factor for AD [106]. Secondary transcriptional and epigenetic changes related to the immune response have also been detected in postmortem brain of AD patients. Increased microglial transcriptomic signatures in AD brains were found to correspond to both altered cell-type composition and microglia activation [107]. Interestingly, some AD loci identified by GWAS and related to immune function overlapped orthologous enhancers with increased activity in a mouse model of AD [107], suggesting a link between primary genetic causes with cellular and molecular events related to AD pathology. Altered inflammatory pathways and immune response in AD makes it reasonable to hypothesize the involvement of infectious agents in triggering or aggravating the disease, and a number of them has been proposed, including viruses, bacteria, and fungi.

Very recent studies have revitalized the idea that a viral infection can contribute to the pathogenicity of Alzheimer’s disease, particularly herpesviruses. The contribution of herpesviruses was proposed almost four decades ago [108] on the basis of a shared predilection for limbic areas in the temporal lobe for both the neuropathological onset of AD and herpesvirus acute encephalitis. Among herpesviruses, HHV-1 attracted most of the attention [109]. HHV-1, which causes HSV encephalitis, was proposed as a putative agent in AD following infection through trigeminal ganglia innervating extra-parenchymal regions close to brain areas of the mesial temporal lobe. In the 1990s, Itzhaki and colleagues found DNA from HHV-1 in brains from both, AD patients and controls [110] and also reported that the presence of HHV-1 DNA was correlated to an increase in the risk for AD when it occurred in combination with carrying the APOE-epsilon4 allele [111]. A decade later, and after a few more attempts to link the presence of HHV-1 DNA with AD, sometimes with contradictory results [112], the application of in situ PCR showed HHV-1 in 90% of Aβ plaques in AD patients [113]. 

The presence of HHV-6 and other human herpesviruses in the brain in association with AD have historically attracted less attention although some are neuropathic and highly prevalent. In addition, and as discussed above, HHV-6 has been associated to encephalitis of limbic predilection [43,114] and, epidemiologically, to epilepsy of the mesial temporal lobe with hippocampal sclerosis, which is in line of Ball’s hypothesis of co-localized damage of AD and herpesviruses. In 2002, Itzhaki lab surveyed the fronto-temporal cortices of patients of AD and controls for the presence of DNA HHV-6, together with Epstein–Barr virus (HHV-4) and cytomegalovirus (HHV-5) [115]. In regard to HHV-6, their PCR analysis showed association between AD and HHV-6 (OR=3.86), based on a total of 85 cases. The same study design applied to patients of vascular dementia (VD) found a smaller, effect (OR = 1.3) in a total of 50 cases. Finally, another study, also testing HHV-1 and HHV-3, found no association between HHV-6 and AD in a total of 65 cases, in which around 80% of brains from both AD and controls were positive for the PCR against the HHV-6 DNA [112]. Puzzlingly, this study found an opposite “protective” effect for HHV-1 in AD. 

Several studies focused on other herpesviruses, producing diverse outcomes. A recent meta-analysis integrated 57 of such studies analyzing the relationship between the infection of different herpesviruses and several types of dementia, and mild cognitive impairment published until 2017 [46]. This meta-analysis also incorporated serological evidences and took into account the type of infection (past or active infection/reactivation) and the type of sample tested (e.g., blood or brain). The study revealed high variability in the results for the eight herpesviruses. Importantly, the assessment of quality of the bulk of evidences showed it was quite low, attending to study designs or risk of biases and detecting data inconsistencies. In the case of HHV-6, the combined effect of the three studied discussed above assessing DNA by PCR in brains of AD and VD yielded a positive OR of 2.47 (95% CI of 1.25 to 4.86) [46]. Additionally, studies comparing HHV-6 DNA from blood or anti-HHV-6 serological measures on serum and intrathecal samples yielded small or no differences between cases of dementia and controls. 

Even if multiple studies suggested an association of herpesvirus to the risk of AD and other types of dementia, conclusions have typically not been granted by statistically robust and replicated observations. The status of this question has been revised as a consequence of a recent large-scale computational analysis of next-generation sequencing data from large cohorts of AD cases [32]. This study started by the unsupervised analysis of microarray data of laser micro dissected neurons from hippocampus and entorhinal cortex in pre-clinical and clinical AD individuals and controls. Molecular network alterations in AD displayed enrichment in gene-sets associated to viral biology, and also pinpointed particular miRNAs, such as miR-155, with described effects on aspects of the biology of multiple herpesviruses, including HHV-6A and EBV. A viral ortholog of this miRNA is also found in the genome of the Kaposi’s sarcoma-associated herpesvirus (KSHV, or HHV-8) [116,117,118]. The authors then set out to investigate differential abundance of viral DNA and RNA in AD on data from four cohorts: Mount Sinai Brain Bank (MSBB), the Religious Order Study (ROS), the Memory and Ageing Project (MAP) and the Mayo Temporal Cortex (MAYO TCX). The analyzed dataset consisted in a total of 1519 samples and included brain regions from the temporal and frontal cortex. The detection of viruses relied on a modified version of the pipeline ViromeScan [119] that in summary compared all non-human mapping reads with a database of species unique 31-mers built from a dataset of 515 reference suspected human viral sequences from the NCBI Viral Genomes Resource [120]. Among all viruses assayed, and considering the meta-analysis of all cohorts, a significant increase in viral RNA in AD samples was found for HHV-6A, HHV-6B, HHV-1, and HHV-7 in temporal and frontal cortex. Specifically, in the MSBB cohort, HHV-6A and HHV-7 were found over-represented in the anterior pre-frontal cortex and superior temporal gyrus and they also observed significant differences, albeit discordant between the two regions, for HHV-6B. The authors also tested for differences at the DNA level using reads derived from whole-exome sequencing in the superior temporal gyrus of the MSBB cohort and confirmed an increase for HHV-6A. Moreover, HHV-6A abundance was correlated with cell-type composition, for example, it correlated negatively with the proportion of neurons in the temporal gyrus, as estimated by cell-type deconvolution using brain single-cell signatures. A final experiment revealed how genetically manipulated mice models with impaired production of miR-155 showed a significant increased β-amyloid plaque density compared to unimpaired mice. Given that HHV-6 infection has been proven to decrease the expression of miR-155 in other cell types [116,121], this result might be suggestive of one of the mechanisms linking HHV-6 to AD.

Even if these important findings are correlative in nature and provide no definitive demonstration of a causative role of HHV-6 in AD, they represented a novel approach, are groundbreaking and put HHV-6 under a more intense focus. Steven Jacobson’s lab has published recently a replicative effort of the Readhead et al. 2018 study [32]. The replication is based on three overlaping cohorts and another one from Johns Hopkins Brain Resource Center (JHBRC), but uses different detection methods and a different computational viral detection pipeline, and focuses only on dorsolateral prefrontal cortex (DLPFC) [44]. Over 900 RNA-seq samples from the ROS, MAP, and MSBB cohorts were screened for multiple pathogens, including 118 human viruses, using a pipeline embeded in the Genome Analisys Toolkit from the Broad Institute called PathSeq [122], using their default reference databases. In addition, Allnutt et al. [44] tested HHV-6 levels in over 700 samples by the highly sensitive method of ddPCR. In this similarly large and powered analysis compared to Readhead et al 2018, however, the authors did not find any significant correlation between HHV-6 PathSeq score, or ddPCR estimated HHV-6 load, and AD diagnose or severity. It is noteworthy, and puzzling, that the number of individuals that showed non-zero detection of HHV-6 was very low in this study, only 6 individuals using PathSeq and 28 by ddPCR, in sharp contrast to Readhead et al. 2018 [32], where more than half of the samples in each group were positive. This differences in detection rates cannot be attributed to the brain region analysed, since it was also observed in those samples derived from DLPFC [32]. It can be argued that the different methods to measure expression of herpesvirus transcripts might account for such dramatic differences in detection rates between these two studies, particulalry considering that latency transcripts are lowly expressed. But what is perhaps more stricking is that the presence of the virus’ DNA is also dramatically lower in Allnutt et al., which is at odds with HHV-6 DNA prevalences reported to date in the brains of healthy individuals tested in multiple studies using a variety of detection technics, probes and tissue preparation protocols (Appendix A). 

Even if the epidemiological relationships between HHV-6, and other herpesviruses, with dementia could be established, the role of these viruses in the pathogenesis of the disease, if any, is yet to be clarified. As for multiple sclerosis and other neurological conditions, the suggested role of herpesvirus in AD ranges from (i) mere bystanders, secondarily reactivated by other pathogenic events related to inflammation in ageing and AD; or (ii) secondary but contributing agents aggravating AD pathology; to (iii) direct causal pathogenic factors. A few studies on cell cultures and animal models has provided some evidence on the mechanisms by which these viruses could be manifesting their effects. Many of these studies has been performed on HHV-1 and have shown cross-talks between HHV-1 and well-known molecular hallmarks in AD: Aβ and hyperphosphorylated Tau [109]. Neural cultures infected by HHV-1 presented Aβ [123] and phospho–tau intracellular accumulation, the later enhanced by the virus-mediated induction of GSK3-beta and PKA [124,125]; those effect could be reduced by anti-viral drugs [126,127]. HHV-1 infection of cultured cortical neurons also showed synaptic [128] and lysosomal [129] impairment. A recent study has notably strengthened the link between HHV-1 as well as the two HHV-6s with the aggregation of Aβ [33], assigning to Aβ a role in entrapping these herpesviruses as a protective mechanism against infection. The authors injected HHV-1 in the hippocampus of WT mice and of mice of an AD model provoking encephalitis. The low homology of HHV-6 entry human receptors in mouse precluded replicating the experimental design in HHV-6, but HHV-6 was tested in neuronal brain organoids presenting Aβ deposition and neurofibrillary tangles. The mouse model of AD showed increased survival to HHV-1 encephalitis, the cell cultures containing Aβ inhibited HHV-1 and HHV-6, and electron microscopy revealed that fibers of Aβ were able to bind herpesviruses [33]. Aβ deposits were present twice as much earlier in organoids infected with HHV-1 and HHV-6 viruses, indicating induction by infection. The colocalization between Aβ fibrils and the viruses suggest a protective role of Aβ by immobilizing virions. Whether Aβ fibrilization is induced by these herpesviruses in vivo and contributes to the pathogenesis of AD is not known, but while HHV-1 DNA has been detected in Aβ plaques in AD [113], HHV-6 antigens have not been detected by immunohistochemistry in AD brains, even in very advanced cases of the diseases and selecting cases with highest evidence of HHV-6 DNA assessed by qPCR (Figure 4). We analyzed the presence of HHV-6 DNA in samples from different Braak stages of AD and controls. We detected through real time PCR very few samples with positive HHV-6, most of which close to the detection limit. We selected a PCR-positive individual with a late stage of AD. We interrogated multiple brain regions by IHQ using an anti-p41 antibody (Abcam) but we only found reactivity in what appear to be corpora amilacea (Figure 4).

Despite strong in vitro studies linking herpesviruses to the amyloid cascade hypothesis, to which extent the putative contribution of HHV-1 or HHV-6 to neurodegeneration occurs in vivo via its relationship with Aβ and thus manifest specifically in AD and VD is not clear. For example, HHV-6A quantification of RNA-seq libraries derived from samples of patients of progressive supranuclear palsy (PSP) also demonstrated an increase compared to controls, even higher than in AD samples [32]. This neurodegenerative disease presents neurofibrillary tangles but not Aβ accumulation. No differences in the presence of HHV-6 and HHV-1 DNA measured by PCR in the brains of patients with Parkinson’s disease compared to controls was reported [112], but one study described HHV-6 staining in neurons in 2 out of 4 PD brains using the p101K antibody [91], whereas the same study failed to detected similarly strong staining in AD samples.

Importantly, the relationship between herpesvirus infection and AD, or other neurodegenerative diseases, might be masked by the tempo of the effects of the infection and the moment of detection, particularly considering the protracted period of time between the disease onset and clinical symptoms of dementia [130]. An example of such putative confounding effect can be illustrated with the case of HHV-3, in where retrospective cohort studies showed increased risk of dementia in individuals with an history of herpes zoster ophthalmicus, caused by reactivation of HHV-3 [109,131], but no differences in postmortem brains or serum antibodies were observed between cases and controls [112,132]. Such type of longitudinal studies contrasting an history of acute HHV-6 infection or reactivation and AD and other types of dementia may shed light on this unsolved question [131]. Patients of cancers that very often have suffered HHV-6 reactivation and sometimes encephalitis, as a consequence of hematopoietic stem cell or umbilical cord blood transplants, are a valuable subgroup of the population to follow up for an increased risk of specific types of dementia. Extrapolating the direct role of HHV-6 in inducing Aβ fibrilization in vitro, we would expect increased incidence in AD in individuals with an history of HHV-6 encephalitis, and particularly among those presenting alterations in hippocampus and mesial temporal lobe, which can be persistent and explain enduring cognitive deficits detected in the follow up of such patients [61].

In conclusion, a complete understanding of the spatiotemporal dynamics of the activity of HHV-6, as well as of other herpesviruses, in the brain in relation to AD onset and progression will be fundamental to discriminate their role in the disease, from causative agents to mere consequences of the changes produced by the disease at later stages, such as exacerbated inflammatory response or accumulated damage in the blood–brain barrier. Longitudinal and comorbidity studies could lead the strategy to identify such relevant interactions with the appearance and/or exacerbation of neurodegenerative processes and refine the research on putative anti-viral therapies. 

## Figures and Tables

**Figure 1 biomolecules-10-01520-f001:**
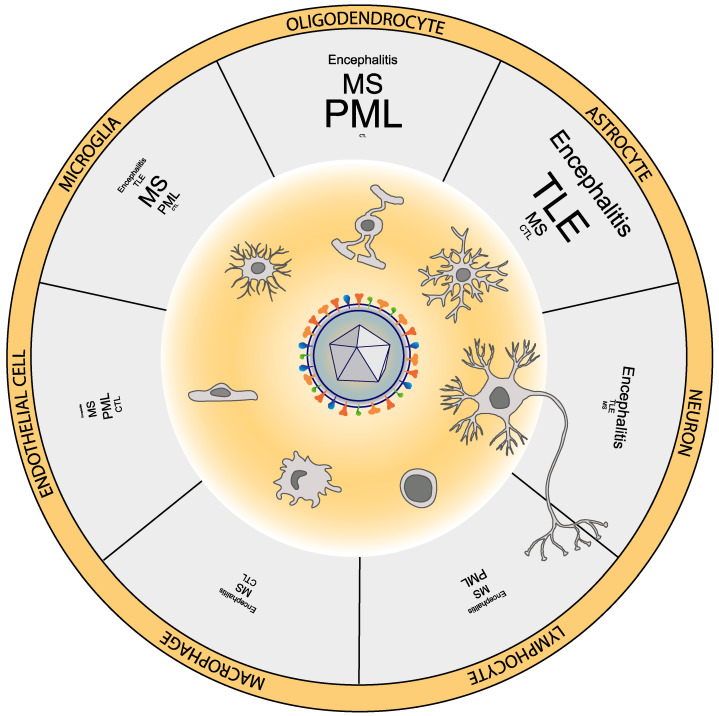
Cell types with positive immunoreactivity against human herpesvirus 6 (HHV-6) antigens across studies. MS: Multiple sclerosis, TLE: Temporal lobe epilepsy, PML: Progressive multifocal encephalopathy. The size of the text indicating each neurological condition is scaled according to the proportion of studies of each condition in which the cell type is detected by immunohistochemistry. For example, most cases of encephalitis or TLE describe staining in astrocytes but almost none in controls (CTL).

**Figure 2 biomolecules-10-01520-f002:**
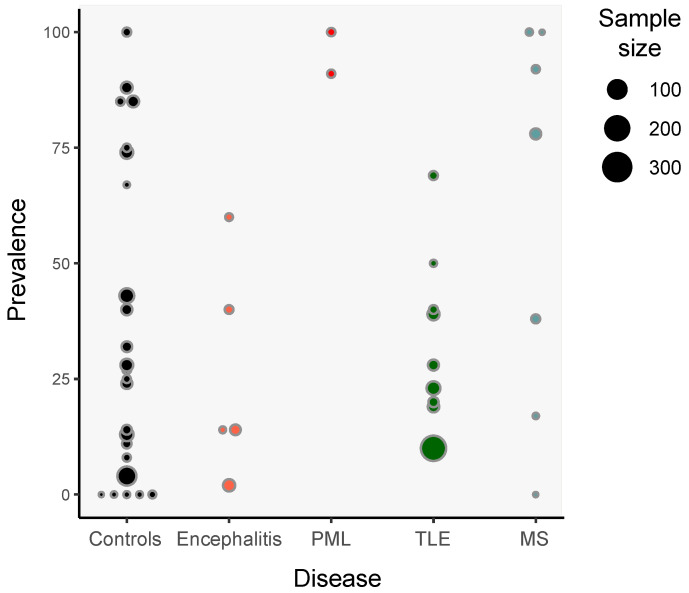
Dotplot indicating the prevalence of detection of HHV-6 DNA in studies with a sample size greater of at least 5 individuals in different neurological conditions and controls with no neurological conditions. The size of each dot is proportional to the sample size of each study. When multiple regions were assessed per individual, we chose the one with the highest HHV-6 prevalence.

**Figure 3 biomolecules-10-01520-f003:**
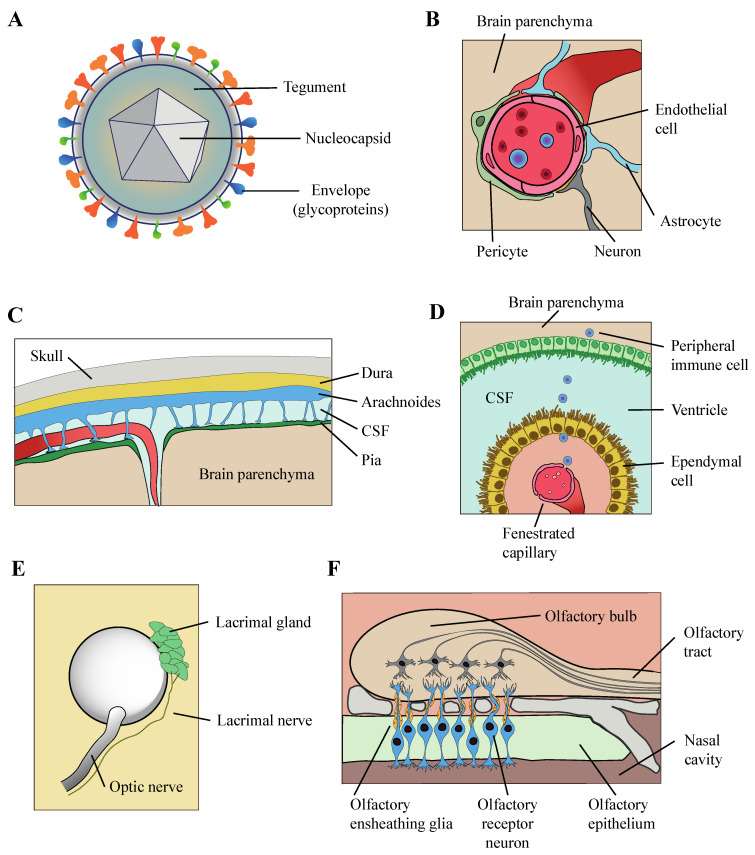
Schematic representation of HHV-6 virion structure (**A**) and suggested HHV-6 entry routes to the brain parenchyma: (**B**) The blood–brain barrier; (**C**) the meningeal barrier; (**D**) the blood–CSF barrier at the choroid plexus; (**E**) optic infection through lacrimal or optic nerves, and (**F**) the olfactory pathway. CSF: Cerebrospinal fluid.

**Figure 4 biomolecules-10-01520-f004:**
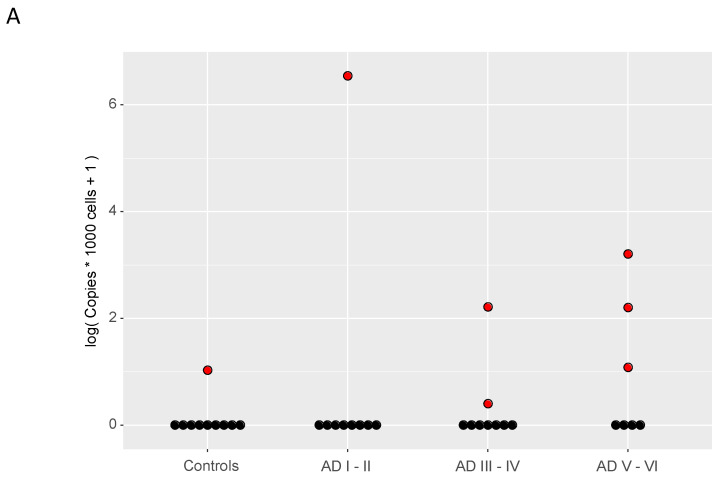
(**A**) qPCR quantification of HHV-6 number of copies per cell in cases of Alzheimer’s disease (AD) at different Braak and Braak stages (I–II, n = 9; III–IV, n = 9; V–VI, n = 7) and controls (n = 10) with primers designed against the U3 region. HHV-6 copy number per cell was determined by comparison with an LCL infected with the congenital form of the virus. The copy number of a reference LCL (NA18999), was determined to be 1 HHV-6 copy per cell in [11] using ddPCR. (**B**) Double-labelling immunofluorescence and confocal microscopy to mouse monoclonal anti-HHV-6 (Abcam ab128404, diluted 1/50; green), and rabbit anti- glial fibrillary acidic protein/GFAP (Diagnostic BioSystems RP014-S, diluted 1/400; red) antibodies. HHV-6 immunoreactivity is only found in the peripheral region of corpora amilacea (arrows) in the periventricular white matter of the temporal lobe in one case of AD stage V-VI and positive PCR. Nuclei are counterstained with DRAQ5^TM^ (blue); bar = 25 μm.

**Table 1 biomolecules-10-01520-t001:** Characteristics of the antibodies used across studies. Blue: MS; Green: TLE; Red: Encephalitis/PML; Black: Others. The color indicates the main focus of the study which may interrogate multiple conditions.

Protein Target	Stage	Localization	Antibody	Type	Species	Source	Ref	Cell Type
p101	Late	Tegument	**C3108-103**	Mouse monoclonal	B	Dr. Philip Pellet (Pellet et al. 1993)	Drobyski et al. 1994 Knox et al. 1995 Challoner et al. 1995 Saito et al. 1995 Friedman et al. 1999	Astrocytes Neurons Oligodendrocytes Macrophague Microglia
						Chemicon	Challoner et al. 1995 Friedman et al. 1999 Mock et al. 1999 Ito et al. 2000 Blumberg et al. 2000 Wainwright et al. 2001 Goodman et al. 2003	Astrocytes Neurons Oligodendrocytes Microglia Endothelial Lymphocytes
						USBiological Cat# H2034-17A	Opsahl et al. 2005	Oligodendrocytes
						Abcam Cat# ab64536	Li et al. 2011	Astrocytes Microglia
						Abcam Cat# ab128404	Liu et al. 2018	Astrocytes Neurons
			Unk	Unk		Biometria	Novoa et al. 1997	Astrocytes Oligodendrocytes Lymphocytes
			Unk	Unk		Virotech	Wang et al. 1999	Microglia Macrophages Endothelial Lymphocytes
Unk	Unk	Unk	#30-HSB	Rabbit antiserum	Unk	Dr. Donald Carrigan (Russler et al. 1991)	Drobyski et al. 1994 Knox et al. 1995 Mckenzie et al. 1995 Carrigan et al. 1996	Astrocytes Neurons Oligodendrocytes Microglia
p41	Early	Nuclear	C5	Mouse monoclonal	A and B	Biodesign (Agulnick et al. 1993)	Challoner et al. 1995	Astrocytes Neurons Oligodendrocytes Macrophague
			Unk	Unk	Unk	Virotech	Mock et al. 1999 Ito et al. 2000 Blumberg et al. 2000 Wainwright et al. 2001 Goodman et al. 2003	Neurons Oligodendrocytes Microglia Endothelial Lymphocytes
			9A5D12	Mouse monoclonal	A and B	Dr. Bala Chandran (Balachandran et al. 1989)	Saito et al. 1995 Wagner et al. 1997	N/A
			Unk	Mouse monoclonal	Unk	Autogen-Bioclear	Opsahl et al. 2005	Oligodendrocytes
			Unk	Unk	Unk	Unk	Cuomo et al. 2001	Glial cells (Schwann)
gp116/gp64/gp54	Late	Core	6A5D5	Mouse monoclonal	A and B	Dr. Bala Chandran	Saito et al. 1995	N/A
			6A6G3	Mouse monoclonal	A and B	Advanced Biotechnologies Cat# 13-219-001 or 13-218-100 (Balachandran et al. 1989)	Goodman et al. 2003 Donati et al. 2003 Fotheringham et al. 2007a Fotheringham et al. 2007b Niehusmann et al. 2010 Esposito et al. 2015	Astrocytes Neurons Oligodendrocytes Macrophages Lymphocytes
						HHV-6 Foundation	Huang et al. 2015	Neurons
			Unk	Mouse monoclonal	A and B	Autogen-Bioclear	Opsahl et al. 2005	Oligodendrocytes
gp82/gp105	Late	Envelope	UK82	Rabbit antiserum	Unk	Dr. Bala Chandran	Saito et al. 1995	N/A
			2D6	Mouse monoclonal	A	Dr. Bala Chandran (Balachandran et al. 1989)	Knox et al. 2000	N/A
gp110/60	Late	Envelope	H-AR3	Mouse monoclonal	Unk	Dr. Luca	Wagner et al. 1997	Astrocytes Neurons Oligodendrocytes
			Unk	Unk	Unk	Unk	Le Guennec et al. 2017	Glial cells
gH	Late	Envelope	OHV-3	Mouse monoclonal	B	Advanced Biotechnologies (Okuno et al. 1990)	Knox et al. 2000	N/A
						HHV-6 Foundation	Huang et al. 2015	Neurons
37KDa early antigen	Early	Unk	1.B.367	Mouse monoclonal	A	USBiologicalCat# H2034-01	Opsahl et al. 2005	Oligodendrocytes
U94	Latency	Nuclear	MORI	Mouse monoclonal	B	HHV-6 Foundation	Huang et al. 2015	Neurons
Unk	Unk	Unk	sc-65463	Mouse monoclonal	A and B	Santa Cruz	Chapenko et al. 2016 Skuja et al. 2017	Astrocytes Oligodendrocytes Microglia Endothelial Lymphocytes Fibroblasts

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
