# Peer review of "The Presence of Human Herpesvirus 6 in the Brain in Health and Disease"

_biomolecules, 2020, doi:10.3390/biom10111520_

Round 1

Reviewer 1 Report

In this review entitled: The presence of Human Herpesvirus 6 in the brain in health and disease, Santpere G. et al. reviewed the detection of Human Herpesvirus 6 ( HHV6) DNA and antigen in multiple neurological conditions. This review is based on a total of 48 studies that investigated the presence of HHV6 DNA and antigens from individuals with neurological conditions, such as encephalitis,  epilepsy, multiple sclerosis, and Alzheimer’s disease (AD). The presence of HHV-6 antigens was well summarized in Fig. 1. This review also included the recent experimental finding of HHV6 in the brains of the AD patient.

This review could improve significantly by including a figure that summarizes cell types with positive DNA detection of HHV-6. It will be easy to follow if the review includes a summary of HHV-6 latency sites, which will explain why HHV-6 was detected in healthy people. Although several neurological conditions were reviewed, the abstract fails to summarize the association of HHV-6 with different neurological conditions.  

Author Response

We thank the reviewer for the suggestion. We have added a new figure summarizing the prevalence of HHV-6 DNA across neurological conditions and sample sizes.

Reviewer 2 Report

The review entitled “The presence of Human Herpesvirus 6 in the brain in health and disease” provided a very detailed and interesting overview of the current knowledge on the entry routes and direct presence of HHV-6 in the brain parenchyma. The results obtained by different groups about the presence of HHV-6 at DNA, RNA, and protein levels in healthy subjects and in patients with neurological conditions, including Alzheimer’s disease are reported.

In my opinion, the topic is relevant, and the manuscript is overall well-written and organized. References are appropriate and updated.

Below my suggestions, that I think would improve the manuscript:

  • Paragraph 1 (Introduction): a more detailed description of the HHV-6 structure should be added, including a schematic picture of the virion;
  • Paragraph 4: in my opinion, a cartoon describing the HHV-6 entry routes would be very useful to help the reader to summarize the topic;
  • Minor: “In vivo” and “in vitro” should be aligned in italics.

Author Response

Below my suggestions, that I think would improve the manuscript:

  • Paragraph 1 (Introduction): a more detailed description of the HHV-6 structure should be added, including a schematic picture of the virion;
  • Paragraph 4: in my opinion, a cartoon describing the HHV-6 entry routes would be very useful to help the reader to summarize the topic;

We thank the reviewer for the suggestion. We created a new figure featuring both HHV-6 virion structure and suggested routes of entry into the brain parenchyma.

  • Minor: “In vivo” and “in vitro” should be aligned in italics.

Amended

Reviewer 3 Report

The manuscript by Gabriel Santpere et al. reviews knowledge on human herpesvirus 6-A and -B infecting almost the entire worldwide population in childhood and on the viral entry routes playing a role in the etiology of neurodegenerative diseases. In the manuscript titled “The presence of Human Herpesvirus 6 in the brain in health and disease”, authors provide a balanced overview on this topic describing the presence of HHV-6 in the brain at the level of DNA, RNA and proteins. This manuscript is aimed to summarize current information on (1) HHV-6 cell tropism in both non-pathological and various neurological conditions such as encephalitis, epilepsy, multiple sclerosis and Alzheimer’s disease and (2) HHV-6 detection methods. This review summarizes information on the most frequently used antibodies for detection of virus proteins by immunohistochemistry. It contains references to 131 different sources of literature, providing a sufficient overview of the topic. The manuscript has an important clinical significance of research on putative antiviral therapies, and would be of significant interest to the readers.

Minor comments:

Line 118: something is missing (it seems that the sentence is broken by Figure).

Line 490: The legend for figure 2 should be clarified (figure 2B is not easy to follow without the informative description in the legend). There are 3 photos which should be described. Is the rabbit anti-GFAP green? Does the area shown pertain to gray or white matter? Why was antibody for HHV-6B used? Why an antibody, which recognizes late antigen, was not used? This information should be added in Line 527 as well.

Line 491: “oligos” should be clarified.

References should be corrected (for example: Line 405 and Line 460; Line 738 and Line 776: journal names and pages are missing; Line 860 and Line 911: years, pages are missing).

Author Response

Minor comments:

Line 118: something is missing (it seems that the sentence is broken by Figure).

Line 490: The legend for figure 2 should be clarified (figure 2B is not easy to follow without the informative description in the legend). There are 3 photos which should be described. Is the rabbit anti-GFAP green? Does the area shown pertain to gray or white matter? Why was antibody for HHV-6B used? Why an antibody, which recognizes late antigen, was not used? This information should be added in Line 527 as well.

Line 491: “oligos” should be clarified.

References should be corrected (for example: Line 405 and Line 460; Line 738 and Line 776: journal names and pages are missing; Line 860 and Line 911: years, pages are missing).

We amended all minor comments.

Reviewer 4 Report

In the manuscript "The presence of Human Herpesvirus 6 in the brain in health and disease" G Santpere, M Telford, P Andrés-Benito, A Navarro and I Ferrer focused on the presence of HHV-6 in the brain resulted from nervous system invasion after an horizontal infection. They reviewed current knowledge about the entry routes and virus presence in the brain parenchyma in healthy subjects and in those with neurological conditions including patients with Alzheimer’s disease.

This MS is very well written, fluent and pleasant to read. The narrative is simple and guide the reader through the logic of the topic.

Just some minor revisions are required, here some as example:

Line 99: “Readhead et al. 2018, Allnut et al. 2020” the ref must be conformed 

Line 126-127: a bracket miss.

Line 328: “… in and additional”, maybe the correct is “in an additional”?

Line 360: “(Dietrich et al. 2004)”, please conform

Line 409: “(Fotheringham et al. 2007, Akhyani et al. 2007)” again, please conform

Line 416: “…HSV-1 and HSV-3”, for HHV1 be consistent, regarding HSV-3 based on the ref they cited, I suppose authors meant HHV-3 (VZV).

Line 440: ”Kapopsi's”, please amend

Line 460: “Caselli et al. 2017, Rizzo et al. 2017)”

Line 494: Plese insert the number of ref for Telford et al. 2018.

Line 496-497: Authors reported that both antibodies are green, please amend.

Figure 2 (A) caption, what about the difference in the copies? Any significance? Authors should report also the information about stats in the caption (i.e., t-test, ANOVA and relative post hoc, etc) and in case of significance indicate with symbols.

Author Response

Just some minor revisions are required, here some as example:

Line 99: “Readhead et al. 2018, Allnut et al. 2020” the ref must be conformed 

Line 126-127: a bracket miss.

Line 328: “… in and additional”, maybe the correct is “in an additional”?

Line 360: “(Dietrich et al. 2004)”, please conform

Line 409: “(Fotheringham et al. 2007, Akhyani et al. 2007)” again, please conform

Line 416: “…HSV-1 and HSV-3”, for HHV1 be consistent, regarding HSV-3 based on the ref they cited, I suppose authors meant HHV-3 (VZV).

Line 440: ”Kapopsi's”, please amend

Line 460: “Caselli et al. 2017, Rizzo et al. 2017)”

Line 494: Plese insert the number of ref for Telford et al. 2018.

Line 496-497: Authors reported that both antibodies are green, please amend.

We amended all minor comments.

Figure 2 (A) caption, what about the difference in the copies? Any significance? Authors should report also the information about stats in the caption (i.e., t-test, ANOVA and relative post hoc, etc) and in case of significance indicate with symbols.

We tested the possible differences in viral load using ANOVA and found no differences (p=0.45). Of note, we only used that analysis to select cases for immunohistochemistry.